# Understanding Information Processing and Protective Behaviors during the Pandemic: A Three-Wave Longitudinal Study

**DOI:** 10.3390/ijerph20054041

**Published:** 2023-02-24

**Authors:** Weidan Cao, Qinghua Yang, Xinyao Zhang

**Affiliations:** 1Department of Strategic Communication, Edward R. Murrow College of Communication, Washington State University, Pullman, WA 99164, USA; 2College of Communication, Texas Christian University, Fort Worth, TX 76129, USA; 3School of Public Health, Jilin University, Changchun 130012, China

**Keywords:** systematic information processing, risk information seeking and processing model, pandemic, protective behaviors

## Abstract

Background: Few existing studies have examined information processing as an independent variable to predict subsequent information behaviors in a pandemic context, and the mechanism of subsequent information behavior processing following the initial or prior information behavior is unclear. Objective: Our study aims to apply the risk information seeking and processing model to explain the mechanism of subsequent systematic information processing in the context of the COVID-19 pandemic. Methods: A three-wave longitudinal online national survey was administered during the period of July 2020 to September 2020. Path analysis was conducted to test the relationships between prior and subsequent systematic information processing and protective behaviors. Results: One important finding was the key role of prior systematic information processing, as indirect hazard experience was found to be a direct predictor of risk perception (*β* = 0.15, *p* = 0.004) and an indirect predictor of protective behaviors. Another important finding was the central role of information insufficiency as a mediator/driving force in subsequent systematic information processing and protective behavior. Conclusions: The study has made important contributions in that it extends the scholarship on health information behaviors by (a) highlighting that relevant hazard experience in risk information seeking and processing model should be expanded to include indirect experience, and (b) providing the mechanism of subsequent systematic information processing following prior information processing. Our study also provides practical implications on health/risk communication and protective behaviors’ promotion in the pandemic context.

## 1. Introduction

Information behavior refers to information users’ active or passive information seeking, processing, or management [1]. Theoretical models of information behavior, such as the risk information seeking and processing (RISP) model [2] and the planned risk information seeking model [3], have been developed to systematically explain the predictors and/or mediators (e.g., cognitive or emotional factors) of information behavior and the consequences of information behavior (e.g., health behavior change). Those theoretical models have been extensively tested to provide empirical support for the mechanisms. In those empirical studies, the information behavior has mainly been tested, consistent with the aforementioned theories, as an outcome or a dependent variable only, in a pandemic context.

However, in a pandemic context, initial information behavior, can also be viewed as an exogenous variable that may predict future information behavior since the information behavior is an ongoing dynamic process that has different stages [4]. As demonstrated in Wilson’s information behavior model [5], a prior information behavior will lead to a person’s evaluation of whether or not more information is needed to make decisions. If information obtained from the initial information behavior is insufficient, then future information behavior might be warranted. Similarly, information processing has different stages. Specifically, the initial/prior stage serves as the process of gaining indirect experience of the situation/hazard that predicts cognitive and affective variables, and predicts subsequent information processing. Then, the subsequent information processing directly predicts behavior change. To the best of our knowledge, however, few existing studies have employed information processing as an independent variable in a pandemic context, and therefore, the mechanism of the ongoing/subsequent information behavior process is unclear. In addition, there is a lack of evidence to show whether or not the existing theoretical frameworks are still applicable to this process of subsequent information behavior following the initial or prior information behavior.

The outbreak of the coronavirus disease 2019 (COVID-19) in China, an infectious disease caused by a novel virus called SARS-CoV-2, provides an optimal context to examine the process of subsequent information behavior due to several reasons. First, people rely heavily on adequate information, especially online information, during emergency situations to make decisions and perform preventive behaviors [6]. Second, given that the information about the prevention and treatment of COVID-19 as a relatively new disease has been updated frequently as the pandemic evolves, people may need to update their knowledge regularly through an information behavior such as information processing. Therefore, information processing can be viewed as an ongoing process and both prior and subsequent information processing behaviors warrant close examination. Finally, since China is the first country that experienced the outbreak of the pandemic, those who were involved in the pandemic had gone through the process from the initial stage of very limited information about the disease to the stage of more abundant information about disease prevention and treatment within the first year of the outbreak.

Given that few existing studies have examined information processing as an independent variable to predict subsequent information behavior in a pandemic context, we proposed a longitudinal study to address the issue. A longitudinal study design was chosen over a cross-sectional design because it would enable us to conduct lagged analyses to establish temporal order. Applying the RISP model [2], complemented by Wilson’s information behavior model [5], the current study has two-fold objectives, namely (a) to explain the mechanism of subsequent information processing following prior information processing, and (b) to explore the antecedents (e.g., prior online information processing) of subsequent online information processing and protective behaviors among residents in China in the context of the COVID-19 pandemic. Through the lens of information behavior as an ongoing dynamic process [5], this study will explain the mechanism of the ongoing/subsequent information behavior process, and will also shed light on the applicability of the RISP model [2] to subsequent information processing. The following section introduces the theoretical framework and explains why the two theoretical models, among other theories, were applied to our study. Then, hypotheses have been proposed based on the theoretical framework.

### 1.1. Theoretical Framework and Hypotheses

Two comprehensive frameworks to explain the mechanism of information behavior—the RISP model [2] and Wilson’s [5] information behavior model—were applied to guide the current study. Both models conceptualize information behavior as a continuous process and have employed a comprehensive number of antecedents to information behavior, with information need/sufficiency as a central motive of the information behavior [6]. The RISP model [2] guided most of the hypotheses testing, while Wilson’s model [5] was used to complement. Our hypotheses were mainly based on the RISP model [2] for two reasons. First, the model is comprehensive in that it includes a variety of factors, such as cognitive and affective factors. If a virus is relatively new, people may feel they do not know enough about it, are unsure about the risks, and are fearful of the outbreak due to the virus’ novelty [7]. Therefore, a comprehensive model with many important variables would be better to help understand the information process. Second, the antecedents (e.g., relevant hazard experience, trust in risk management) postulated in the model are unique and applicable to the COVID-19 context. For instance, the effective control of the spread of COVID-19 involves government policies and actions and the trust in government’s risk management will likely influence people’s risk perception, and protective behaviors (please see the section of “Trust in Risk Management” below).

However, the RISP model [2] did not explicitly theorize the relationship between an individual’s prior information behavior and subsequent information behavior in the model [8]. Wilson’s model [5] was employed as a complement because of its specification of the route or relationship between initial information behavior, ensuing information need, and subsequent information behavior. That is, after the initial information behavior, if the seeker perceived the need for more information, then subsequent information is warranted. Key constructs of the RISP model [2] were examined in our study, including risk perception, relevant hazard experience, affect, knowledge insufficiency, and subjective norms. There were two criteria used to include variables in the study: 1. the constructs are considered central to the RISP model (e.g., risk perception); and 2. the constructs (e.g., trust in risk management) were applicable to China’s pandemic context. Detailed explanations on the RISP constructs [2] and Wilson’s model [5] variables included in our study and on hypotheses developed from the theoretical models are presented below.

### 1.2. Risk Perception and Relevant Hazard Experience

As one of the central constructs of RISP [2], risk perception is influenced by relevant hazard experience, predicts other variables (e.g., affect and information seeking [2,9]), and is conceptualized as one’s perception of the severity and susceptibility of a disease [10,11]. Relevant hazard experience, defined as the personal experience of the same hazard or a similar hazard [2], has been theorized as direct personal experience in the RISP model [2]. However, relevant hazard experience should be expanded to include indirect experience with a hazard or health risk [12,13], especially during the pandemic when direct experience is restricted to some extent because of a variety of reasons (e.g., lockdown, social distancing, travel restriction). Indirect experience or vicarious experience [14] refers to the experience one learns from others’ direct experience, for instance, information drawn from secondary sources [12]. Given that the RISP model did not include indirect experience, although highly relevant to subsequent information behavior in the pandemic context, testing the extended RISP model [2] with expanded operationalization of relevant hazard experience (i.e., both direct and indirect experience) is warranted.

### 1.3. Previous Systematic Processing as Indirect Hazard Experience

Information processing as a type of information behavior can be further categorized into systematic and heuristic processing [2], which are two key terms of dual-process theories [15]. Systematic processing refers to processing information in an effortful and systematic way, whereas heuristic processing refers to processing information in an effortless and superficial way [16,17]. The two modes of information processing predict risk perception in different ways. Specifically, individuals’ risk perception is correlated positively with systematic processing but negatively with heuristic processing [11,18].

Systematic processing focuses on in-depth consideration of the content of messages and can be viewed as a process of vicarious learning [14] or indirect learning. People systematically process information online to obtain a sense of how COVID-positive patients feel about the disease and learn the mitigation behaviors without having to personally contract the disease [12]. During the pandemic, especially when lockdown measures were enforced, the internet served as an important information source [19]. Therefore, the systematic processing of online information about COVID-19, which can be conceptualized as the process of gaining indirect hazard experience [12], is the focus of our study.

### 1.4. Direct Experience, Indirect Experience, and Risk Perception

The RISP model [2] posits that direct experience predicts risk perception. Studies demonstrated that indirect/online experience with COVID-19 predicted protective/preventive behaviors (e.g., restricting non-essential travel, wearing face masks in public) [13,20]. In addition, both direct and indirect experience have been documented as positive predictors of risk perception. For instance, an extensive amount of COVID-19 information obtained from different sources such as social media predicts risk perception (e.g., perceived susceptibility and severity) [21]. Other studies found the indirect effect of social media exposure on preventative behaviors through risk perception [22]. The relationship between systematic information processing as indirect experience and risk perception in the context of COVID-19 can be hypothesized based on the positive relationship between systematic information processing and risk perception in other contexts demonstrated in existing literature (e.g., [23]). The following hypotheses (Hs) were proposed:

**H1.** *Direct experience will positively predict risk perception*.

**H2.** *Prior systematic information processing behavior as indirect experience will positively predict risk perception*.

### 1.5. Trust in Risk Management

Trust has played an important role during the pandemic because it may influence people’s risk perception and their behaviors in response to the pandemic (e.g., physical distancing) [24,25]. General trust is conceptualized as “the belief that most people are trustworthy most of the time and it is related to how much someone trusts people whom one meets for the first time” [25]. In contrast, different types of specific trust (e.g., trust in the government during a pandemic) are trust in specific objects and/or entities [26] and should be distinguished from general trust, since their effects on risk perceptions and behaviors are different [25]. Trust in risk management, a specific type of trust, defined as “A judgment of the amount of trust the respondent has in the ability of others to prevent the respondent from coming to harm” [2], is a predictor of risk perception. Given that such speed and scale of COVID-19 response strategies need the involvement of government [24] and pandemic management strategies and have been implemented by governments around the world [27], trust in the risk management of the government or public officials becomes an important issue. Trust in government action to control the pandemic was listed as a reason for compliance with protective behaviors [28]. In another study, trust in public officials predicted the likelihood of adopting prevention practices [29]. The local Chinese government often performs tasks such as conducting risk management, providing health recommendations, and providing guidelines in response to a pandemic. Although it is an important factor, no study has yet systematically examined the role of trust in government in a pandemic, especially the relationship between trust in government and risk perception, to the best of our knowledge. To fill this gap, H3 was proposed, based on the relationship specified in the RISP model:

**H3.** *Higher levels of trust in local government’s risk management predict lower risk perception*.

### 1.6. Affect

Negative affective responses (e.g., worry or fear) [2,11] are outcomes of risk perception and subsequently predict information behaviors (e.g., information processing) [2]. That is, when people perceive higher levels of susceptibility and/or severity of the pandemic, they are more likely to feel worried or fearful; those affective responses are usually unpleasant and will therefore motivate people to seek and systematically process more pandemic-related health information and then engage in protective behaviors to reduce such negative affect. Many studies have examined the relationship between risk perception and affect (e.g., [7,30,31]). For instance, in a study examining the information processing during an infectious disease (i.e., Zika) outbreak, perceived susceptibility and perceived severity were identified as positive predictors of negative affective responses [7]. Therefore, the following hypotheses were proposed.

**H4.** *Risk perception positively predicts negative affective responses*.

**H5.** *Negative affective responses positively predict systematic processing*.

### 1.7. Knowledge Insufficiency

Information insufficiency is the knowledge needed, in addition to the existing knowledge one has [2], or “perceived need for more information” [32]. Both the RISP model [2] and Wilson’s [5] information behavior model theorize information insufficiency as an essential component. According to the RISP model [2], information insufficiency is influenced by affective responses and predicts information behaviors (e.g., systematic processing), which found support in empirical studies [7,8,33]. Furthermore, studies (for the list, see [2]) found that negative affect are linked to systematic processing. In the context of COVID-19, when people are worried about contracting the disease, they are motivated to process information to protect themselves. When people feel they are not equipped with enough adequate knowledge to protect themselves and others, they are also motivated to systematically process more information related to COVID-19. In addition, according to Wilson’s [5] model, prior information processing as indirect experience will predict future information behavior if one finds the information is not enough. As the pandemic caused by the new virus evolves, researchers begin to know more about the disease and its variants, the adequate protective behaviors, and the efficacy of the vaccination and treatment. Our knowledge of COVID-19 is changing constantly and there is always new information and scientific knowledge to be learned. After the initial information processing behavior, one may find there is still more to learn. Therefore, the following hypotheses were proposed.

**H6.** *Negative affect positively predicts knowledge insufficiency*.

**H7.** *Information insufficiency positively predicts systematic information processing*.

**H8.** *Prior systematic processing as indirect experience positively predicts information insufficiency*.

### 1.8. Subjective Norms

According to RISP, informational subjective norms represent “perceived social normative influences motivating the desire for information sufficiency” [2]. If people feel their important others (e.g., family, friends) collectively believe that it is important to systematically process information, they may decide to systematically process information directly [34]; and/or based on the pressure of social influence for systematic processing, they may perceive the need for more information, which then predicts systematic processing of information [2]. The motivations to realize others’ anticipations and desires in order to be socially acceptable might drive systematic processing [35,36]. Studies have also demonstrated that informational subjective norms directly predicted systematic processing [7]. Moreover, specific behaviors are outcomes of information behaviors [37] and after systematically processing COVID-related health information, people may adopt protective behaviors such as wearing a face mask. Studies have also found that information behaviors are often associated with protective behaviors (e.g., [38]). Therefore, the following hypotheses were proposed.

**H9.** *Systematic processing subjective norms positively predict information insufficiency*.

**H10.** *Systematic processing subjective norms positively predict systematic information processing*.

**H11.** *Systematic information processing positively predicts protective behaviors*.

The model including the hypotheses is presented in Figure 1. In addition to investigate the mechanism of information processing at different stages, we will explore the indirect effects with the research question (RQ) proposed below:

RQ: What is the indirect effect between (a) direct experience, (b) indirect experience, and (c) trust in risk management and protective behaviors?

## 2. Materials and Methods

### 2.1. Participants and Procedure

This study was approved by the Institutional Review Board at a university in China. Participants were adults (18 and older) who resided in mainland China since the start of the pandemic. A three-wave longitudinal online national survey was administered among potential participants by a professional survey company (i.e., Wenjuanxing) during the period of July 2020 to September 2020. The decision to start the survey on July 2020 was based on two facts: 1. By the end of June, COVID-19 had rapidly spread to other countries and become a global pandemic [39]. 2. On 26 June 2020, the ACT accelerator called for $31.1 billion to be invested for COVID-19 diagnoses, treatment research, and vaccination development [40]. Therefore, starting from July 2020, there could be much richer information on COVID-19, such as updated prevention guidelines, news on vaccination development, from many different countries. The longitudinal design enables us to conduct lagged analyses and is appropriate to examine the relationship between prior systematic processing and subsequent systematic processing. The consent form was displayed at the beginning of the survey. By starting the survey, the participants agreed to participate in the study. The participants who finished the survey at time 1 (T1) were contacted to fill out the survey at T2, and those who finished T2 survey were qualified for the survey at T3. The interval between two surveys was two weeks, which was chosen based on the rationale expressed in [38]. To be more specific, factors such as the dropout rate of a longitudinal study, and the potential time needed to initiate protective behaviors after information behaviors were considered to reduce attrition rate while simultaneously allowing ample time for the outcome (i.e., protective behavior) of information seeking behavior to happen [38]. The two-week period is enough for relevant behavior change because of several reasons: 1. First, the adoption and change of information seeking or processing behaviors and protective behaviors (e.g., “maintain 1.5–2.0 m apart when with others outside”) would be relatively easy (compared to smoking or hazardous drinking). 2. In this new pandemic context, behavior changes are anticipated by people in general, because, given the early death and severity rate of the new disease, people will need the latest information to act accordingly to protect themselves and their family members. 3. Facilitating conditions of information seeking behaviors, such as smartphone and Internet access would not be an issue. About 73% of China’s population had subscribed to mobile Internet services as of the end of 2021 [41].

We collected 871 responses at T1, 613 at T2, and 437 at T3, respectively. The participants with three waves of 100% completed surveys and two correct attention check [42] items were included for data analysis. After data cleaning, the final sample size was 365, on which we reported the results.

### 2.2. Measures

For the longitudinal survey, participants’ sociodemographic variables (i.e., age, gender, health, and education levels) were measured at T1. RISP constructs and protective behaviors were measured at T1, and/or T2, and/or T3. The original questionnaire was in English and was translated to Chinese. The Chinese version was then back translated into English and was pretested before survey administration to ensure the clarity and readability of the questionnaire. For the pretest, we invited 10 Chinese native speakers to fill out the Chinese version of the questionnaire. Then, we asked them about the clarity and readability of the questionnaire and made changes accordingly. We then invited another five Chinese native speakers to fill out the revised questionnaire, and no one reported any clarity or readability issues.

### 2.3. RISP Constructs

Direct experience was measured at T1 through two items adapted from existing studies [43,44,45]. Sample questions include asking respondents to rate how frequently they experienced a pandemic (e.g., SARS, H7N9 avian influenza) in the past on a 5-point scale. Trust in risk management (T1 Cronbach’s α = 0.69) was measured at T1 by three items adapted from [46] on a 5-point scale. Sample items included “The local government protects people like me from these risks.” Risk perception (T1 α = 0.80), including perceived susceptibility and perceived severity, adapted from [3,7], was measured at T1 via four items on a 5-point scale. Sample items include “How serious are current COVID-19 threats to your health?”. Affect, including worry and scare, adapted from [3], was measured at T1 via two items on a 5-point scale. Sample items include “Current COVID-19 risks to my health are scary”. Knowledge insufficiency, adapted from [7,47], was measured at T1 via one item (i.e., Think of that same 0 to 100 scale again. Estimate how much knowledge you need to deal adequately with risks to your health). Subjective norms of systematic processing (T1 α = 0.77), adapted from [47], were measured at T1 using four items on a 5-point scale. Sample items include “My family and friends expect me to systematically process information about COVID-19”. Systematic information processing (T2 α = 0.72), adapted from [7], based on [2], was measured at T1 and T2 using five items on a 5-point scale. Sample items include “After I encounter information about COVID-19, I am likely to stop and think about it”. Indirect experience (T1 α = 0.73) was the first wave of systematic information processing.

### 2.4. Protective Behaviors

Protective behaviors (T2 α = 0.70; T3 α = 0.73), adapted from an existing study [48] as well as based on the COVID-19 guidelines [49], were measured at T2 and T3 using four items on a 5-point scale. Sample items include “I maintain 1.5–2.0 m apart when with others outside.”

### 2.5. Analyses

To test the direct effects, path analysis was used to test the constructs (average scores were computed for each construct) of RISP and protective behaviors, with systematic processing and protective behaviors at T1 or T2 controlled in the analyses. Statistical software Mplus 8.0 [50] was employed to test the relationships. In order to test indirect effects using path analysis, direct experience, indirect experience, and trust in risk management at T1 were exogenous variables, risk perception, subjective norms, affect, knowledge insufficiency, and systematic information processing at T2 were mediators, and protective behavior at T3 was specified as an endogenous variable. Stability paths from all T1 variables to the same T2 variables, and from all T2 variables to the same T3 variables were included in the model to perform lagged analyses. Bootstrapping technique, with the number of iterations being set to 5000, was used to obtain bias-corrected 95% confidence to test the indirect effect [51]. According to Hu and Bentler [52], a comparative fit index (CFI) ≥ 0.95, a Tucker–Lewis index (TLI) ≥ 0.95, a root mean square error of approximation (RMSEA) ≤ 0.06, and a standardized root mean square residual (SRMR) < 0.08 indicate good fit of model.

## 3. Results

### 3.1. Descriptive Statistics

The mean age of the participants was 33.81 (Min = 19, Max = 73, SD = 10.64), and 46% participants were female. The majority (67.90%) of the participants had a bachelor’s degree. Please see Table 1 for detailed descriptive statistics.

### 3.2. Model Fit and Hypotheses Testing

The results for the overall fit of proposed model (χ^2^ = 71.44, df = 49, *p* = 0.02, RMSEA = 0.035, CFI = 0.96, TLI = 0.95, SRMR = 0.06) indicated good fit to the data; even the *p* value was significant. Direct experience (*β* = 0.30, *p* < 0.001), indirect experience (*β* = 0.15, *p* = 0.004), but not trust in risk management (*β* = −0.06, *p* = 0.23), significantly predicted risk perception, which supported H1 and H2 but not H3. That risk perception significantly predicted affective responses (*β* = 0.54, *p* < 0.001) at T1 supported H4. Affect (*β* = 0.15, *p* = 0.009), indirect experience (*β* = 0.33, *p* < 0.001), but not systematic processing subjective norms (*β* = 0.11, *p* = 0.08), significantly predicted information insufficiency at T1, which supported H6 and H8 but not H9. Information insufficiency (*β* = 0.14, *p* = 0.004), and subjective norms (*β* = 0.20, *p* < 0.001), but not affect (*β* = −0.04, *p* = 0.46) at T1, significantly predicted systematic information processing at T2 after controlling for T1 systematic information processing; therefore, H7 and H10 were supported, but H5 was not supported. Systematic information processing (*β* = 0.23, *p* < 0.001) at T2 significantly predicted protective behavior at T3 after controlling for T2 protective behaviors; thus, H11 was supported. Please see Figure 2 for the detailed results.

### 3.3. Indirect Effects

Two significant indirect paths from indirect experience to protective behaviors were identified through (a) systematic processing (ES = 0.126, 95% CIs [0.055, 0.191]), and through (b) insufficiency and systematic processing as serial mediators (ES = 0.011, 95% CIs [0.002, 0.026]). All other indirect effects were not significant (see Table 2 for detailed results including effect sizes and confidence intervals). The RQ was answered.

## 4. Discussion

### 4.1. Principal Findings

In this study, guided by the RISP model [2] and Wilson’s model [5], we examined the predictors (e.g., prior systematic processing as indirect experience) of systematic processing and protective behaviors using a three-wave longitudinal survey. Congruent with previous studies [7], the current study demonstrated a significant direct relationship between variables (e.g., from risk perception to affect, from affect to knowledge insufficiency, and from norms to systematic processing).

The most important finding of the study was the key role of prior systematic information processing as indirect experience was found to be a direct predictor of risk perception and an indirect predictor of protective behaviors. Although previous studies found online COVID-19 information exposure was related to preventative behaviors [20], the mechanism of the relationship is unclear. Our study’s result (i.e., prior systematic information processing indirectly predicted protective behavior through a set of mediators) provided a possible explanation of the mechanism: people may systematically process the information through information exposure, which leads to information insufficiency perception and subsequent systematic information processing, and then leads to protective behaviors.

Another important finding of the study was the central role of information insufficiency as a mediator/driving force in subsequent systematic information processing and protective behavior. Previous studies have demonstrated the importance of risk perception, affect, knowledge insufficiency, and subjective norms, as mediators of information behaviors based on cross-sectional analyses [7], but not as mediators in longitudinal studies; in addition, previous studies did not test the relationship between prior systematic processing and subsequent systematic processing. Our study’s result (i.e., indirect effect: indirect experience to insufficiency to systematic processing to protective behavior) demonstrated the essential role of information sufficiency in subsequent systematic processing and suggested that the mechanism of prior systematic processing process may be different from subsequent systematic processing process. For initial systematic processing, people with little knowledge about the disease feel fearful or worried about the disease and those factors (e.g., affect, and information insufficiency) are the driving force of the information behavior, whereas the driving force for subsequent processing is information insufficiency because they already accumulated some but not enough knowledge through the previous round of the information behavior experience. As people gain more experience in information processing through subsequent information behaviors, the factors (e.g., affect, social norms) are no longer as important as they are in the initial information processing, and the existing limited knowledge about COVID-19 will motivate people to know more about the disease and then continue follow-up information behaviors. Even if people still hold negative affect towards COVID-19, it is unlikely they will continue further information processing if they think they possess sufficient information about COVID-19.

Trust in risk management, a variable highly relevant to the pandemic context and an understudied variable, however, did not significantly predict risk perception directly (H3), nor did it predict protective behaviors indirectly in our model. Our interpretation is that trust in risk management can be an important predictor of initial information behaviors, but our study focused on explaining subsequent information behaviors. Trust in risk management may not be a driving force of subsequent information behaviors. Alternatively, it might be a key predicting factor for subsequent information behaviors, but it lacked variability in our data and people’s trust in risk management during COVID-19 was all relatively high in China. Therefore, more studies are needed to test the role of trust in risk management. The fact that H5 (from affect to systematic processing) was not supported but H7 (from knowledge insufficiency to systematic processing) was supported indicated the aforementioned central role of information insufficiency. The fact that H9 (from norms to knowledge insufficiency) was not supported but H8 (from indirect experience to knowledge insufficiency) was supported indicated the aforementioned key role of prior systematic information processing.

### 4.2. Theoretical and Practical Implications

The findings of the current study advance the theoretical frameworks from three perspectives. First, relevant hazard experience, as a construct of RISP [2], should be expanded to conceptually include both direct and indirect experience. Although there are many ways to obtain indirect experience, indirect experience can be operationalized as systematic information processing online, especially given the pandemic context where confinement is recommended for a long period of time. Second, as indicated by Wilson’s model [5] and the results of our study, a feedback loop should be included in the RISP model [2] to highlight the ongoing feature of information behaviors. Considering that information behaviors are an ongoing process, a potential feedback loop is from information behaviors to information insufficiency. That is, if prior information behavior leads to the perception of information insufficiency, future information behavior may be warranted. Third, the mechanism of subsequent systematic information processing can be different from that of prior systematic information processing in that the mediators of the processes can be different; for subsequent information processing, information insufficiency, as a mediator, plays the key role in determining future information behaviors.

The findings also have practical implications. With the goal of protecting people from contracting COVID-19 and exercising protective behaviors, measures and strategies that emphasize the importance of ongoing systematic information processing and help motivate people to systematically process online information from credible sources should be promoted. For instance, for credible health information websites (e.g., CDC, WebMD), a section on the importance and the potential benefits of ongoing systematic information processing can be provided. Given that mass media were indicated as the dominant information source for the pandemic [53], the importance of ongoing systematic information processing can be highlighted on television news and online newspapers. Another important practical implication is highlighting the possibility of information insufficiency, given the ever-changing and evolving nature of the new pandemic, when communicating COVID-19 information to the public. Practically, health websites can communicate the possibility of information insufficiency to its readers by providing a highlighted note at the beginning of the key health information section, despite the disparities in online COVID-19 information seeking [54]. For instance, “Note: Please remember to come back regularly to check for most up-to-date health information, given COVID-19 is a new disease and information on prevention and treatment will be updated frequently as the pandemic evolves.” In addition, official and trusted news sources can include experts and scientists’ perspectives to help disseminate updated scientific knowledge and avoid the spread of misinformation [55]. Thus, continuous and ongoing information behaviors such as systematic processing should be encouraged, which was found to motivate protective behaviors.

### 4.3. Limitations and Future Research

Our study has several limitations. First, the study was conducted using non-probability sampling, although we used a professional survey company to collect data nationally. Therefore, the sample might not be representative of the population. Future studies could replicate the current study using probability sampling. Second, the RISP model [2] has included other important constructs (e.g., perceived channel beliefs and information gathering capacity) that were not examined in this study. Future studies could test the role of those variables in subsequent information behaviors. In addition, studies (e.g., [56]) have indicated that people’s socioeconomic status (SES) is related to the adoption of preventive behaviors. Therefore, in future studies, we will examine the role of demographic factors (e.g., SES) in the process of ongoing systematic information seeking and preventive behaviors.

## 5. Conclusions

In sum, guided by the RISP model [2] and Wilson’s [5] model, this study is among the first endeavors to examine the role of prior systematic information processing as indirect experience in predicting subsequent systematic information processing and protective behaviors using a longitudinal design. The study has made unique contributions in that it extends the scholarship on health information behaviors in two ways. First, the results of the study highlight that the conceptualization of relevant hazard experience in the RISP model [2] should be expanded to include both direct and indirect experience, rather than just focusing on direct experience. Second, our study has provided the mechanism to explain subsequent systematic information processing in which information insufficiency plays an important role. Our study also provides practical implications on health/risk communication and protective behaviors’ promotion in the pandemic context.

## Figures and Tables

**Figure 1 ijerph-20-04041-f001:**
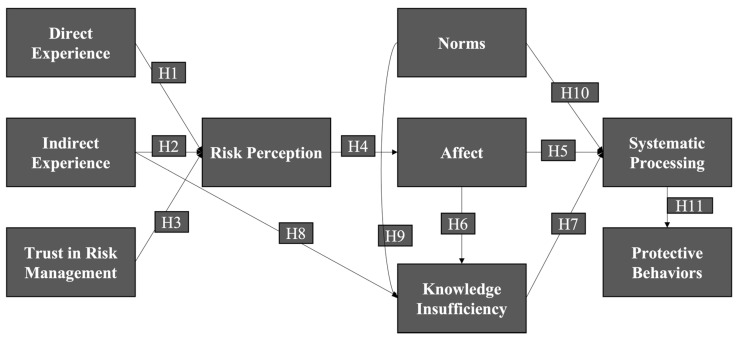
COVID-19 information processing model.

**Figure 2 ijerph-20-04041-f002:**
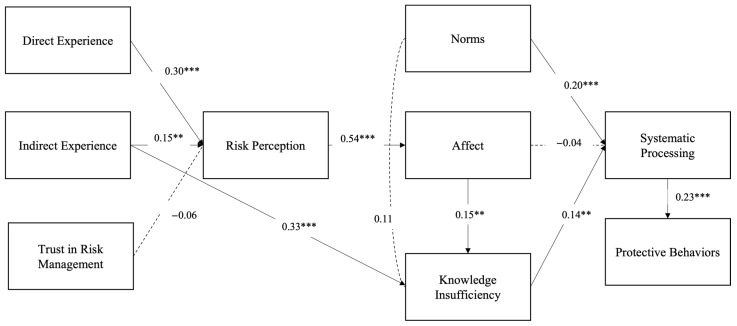
Model with hypotheses testing. ** *p* < 0.01; *** *p* < 0.001.

**Table 1 ijerph-20-04041-t001:** Descriptive statistics of samples at wave 1, wave 2, and wave 3 (N = 365).

	Wave 1	Wave 2	Wave 3
Age (years; M + SD)	33.81 + 10.64		
Female (%)	46.00		
Education (%)			
Below high school	3.60		
Technical or specialized school	2.70		
High school	2.70		
Associate degree	11.00		
Bachelor’s degree	67.90		
Master’s degree or above	12.10		
Health (%)			
Very poor	0.30		
Poor	2.50		
Fair	15.60		
Good	64.70		
Very good	17.00		
Household annual income (RMB) (%)			
<20,000	2.50		
20,001–50,000	9.90		
50,000–100,000	18.10		
100,001–200,000	37.80		
200,001–500,000	27.90		
>500,000	3.80		
Direct experience (M + SD)	3.17 + 1.09		
Indirect experience (M + SD)	4.11 + 0.53		
Trust in risk management (M + SD)	4.44 + 0.52		
Risk perception (M + SD)	2.99 + 0.82		
Affect (M + SD)	3.90 + 0.96		
Knowledge insufficiency (M + SD)	77.83 + 17.20		
Subjective norms (M + SD)	3.72 + 0.76		
Information processing (M + SD)		4.08 + 0.52	
Protective behaviors (M + SD)		3.99 + 0.75	4.04 + 0.78

**Table 2 ijerph-20-04041-t002:** Indirect effects of direct experience, indirect experience, and trust in risk management predicting protective behaviors.

Research Question	Specific Indirect Paths	Mediation	
		Indirect Effect Size	Bias-Corrected Bootstrapping Confidence Intervals
Direct Experience → Protective Behaviors	1. Direct Experience (T1) → Risk Perception (T1) → Systematic Processing (T2) → Protective Behaviors (T3)	−0.003	[−0.01, 0.004]
	2. Direct Experience (T1) → Risk Perception (T1) → Affect (T1) → Systematic Processing (T2) → Protective Behaviors (T3)	−0.001	[−0.005, 0.003]
	3. Direct Experience (T1) → Risk Perception (T1) → Affect (T1) → Insufficiency (T1) → Systematic Processing (T2) → Protective Behaviors (T3)	0.001	[0.000, 0.002]
Indirect Experience → Protective Behaviors	1. Indirect Experience (T1) → Systematic Processing (T2) → Protective Behavior (T3)	0.126	[0.055, 0.191]
	2. Indirect Experience (T1) → Risk Perception (T1) → Systematic Processing (T2) → Protective Behavior (T3)	−0.001	[−0.005, 0.002]
	3. Indirect Experience (T1) → Insufficiency (T1) → Systematic Processing (T2) → Protective Behaviors (T3)	0.011	[0.002, 0.026]
	4. Indirect Experience (T1) → Trust in Risk Management (T1) → Risk Perception (T1) → Systematic Processing (T2) → Protective Behavior (T3)	0.000	[0.000, 0.001]
	5. Indirect Experience (T1) → Risk Perception (T1) → Affect (T1) → Systematic Processing (T2) → Protective Behaviors (T3)	−0.001	[−0.003, 0.001]
	6. Indirect Experience (T1) → Trust in Risk Perception (T1) → Risk Perception (T1) → Affect (T1) → Systematic Processing (T2) → Protective Behaviors (T3)	0.000	[0.000, 0.001]
	7. Indirect Experience (T1) → Risk Perception (T1) → Affect (T1) → Insufficiency (T1) → Systematic Processing (T2) → Protective Behaviors (T3)	0.000	[0.000, 0.001]
	8. Indirect Experience (T1) → Trust in Risk Management (T1) → Risk Perception (T1) → Affect (T1) → Insufficiency (T1) → Systematic Processing (T2) → Protective Behaviors (T3)	0.000	[0.000, 0.000]
Trust in Risk Management → Protective Behaviors	1. Trust in Risk Management (T1) → Risk Perception (T1) → Systematic Processing (T2) → Protective Behaviors	0.001	[−0.002, 0.004]
	2. Trust in Risk Management (T1) → Risk Perception (T1) → Affect (T1) → Systematic Processing (T2) → Protective Behaviors (T3)	0.000	[−0.001, 0.000]
	3. Trust in Risk management (T1) → Risk Perception (T1) → Affect (T1) → Insufficient (T1) → Systematic Processing (T2) → Protective Behaviors (T3)	0.000	[−0.001, 0.000]

## Data Availability

The data presented in this study are available on request from the corresponding author. The data are not publicly available due to participant privacy concerns.

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
