# Peer review of "Understanding Information Processing and Protective Behaviors during the Pandemic: A Three-Wave Longitudinal Study"

_ijerph, 2023, doi:10.3390/ijerph20054041_

Round 1

Reviewer 1 Report

The proposed work is appreciable and the presentation is good but the manuscript still

needs a lot of attention to justify the method proposed in the paper. There are several of

things which are unclear such as novelty, contribution etc. The following are the

suggestions which needs to be incorporated into the work.

(1) Abstract should be restated by adding the importance and contribution of the work.

(2) Introduction section should be completely updated by adding motivation,

organization and novelty of the work. Highlights the importance of proposed method

over the several other existing methods.

(3) Authors should use the common symbols which can be easily understandable and

readable.

(4) Result and discussion section needs to be more generalized ways.

(5)The detail comparison and sensitivity analysis should be added and also the

limitation and advantage should be added

(6) Reference section needs to be polish by adding some recent articles related to

diverse approaches. 

Reviewer 2 Report

The present paper under review deals with the study on Understanding Prior and Subsequent Systematic Information Processing and Protective Behaviors in the Context of COVID-19 Pandemic: A Three-Wave Longitudinal Study. Authors have prepared a well-discussed work which presents a well-considered up-to-date aspect which has interest for all. The novelty of the idea of the authors is obvious. Presented algorithm and analysis are correct. These results may be helpful to the study of this area and other related areas. Therefore, I recommend this article to be accepted for publication, but it requires a minor revision before its acceptance.

1. Throughout my reading, I met some typos. The authors are suggested to check them carefully to improve the quality of the paper.

2. Remove abbreviation from Abstract.

3. In the introduction section, add the motivation and the importance of the considering the present study over the existing studies.

4. The conclusion also needs some enhancements. Clearly state your unique research contributions in the conclusion section and point out some potential directions for future work.

5. Irrelevant references should be deleted from the list of references.

Reviewer 3 Report

Thanks for giving me a chance to review this paper. I have the following suggestions.

Title

·         Concise the title

Introduction

·         Need improvement with new references and remove grammatical mistakes.

Literature review

·         Try to incorporate new literature. Only one reference from the year 2022.

·         Grammar needs improvement.

·         However, you have used two models, but it is better to support and Incorporate the theories based on which you have driven your research model.

Methodology

·         Describe the descriptive statistics adequately.

·         The measurement model is missing.

·         However, you have used SmartPLS, but you have not incorporated any advanced techniques like MGA, IPMA, Q Square

·         The Chinese version was then back translated into English and was pretested before survey administration to ensure the clarity and readability of the questionnaire. Explain in detail the pretested procedure.

·         Also, incorporate the SmartPLS model indicating construct and items.

Discussion

·         The discussion needs to be improved.

·         Why most of the hypothesis are not supported, justify with proper reasoning in the discussion.

Conclusion:

Please add a section of limitation and future work

Reviewer 4 Report

This article is well written, organized correctly and it aims to apply the risk information seeking and processing (RISP) model to explain the mechanism of subsequent systematic information processing in the context of COVID-19 pandemic. The Introduction section with the Materials and Methods section are in a reasonable length, given the premise of the paper. The structure in Results, Discussion and Conclusions parts are well argued, the logic easy to follow and the benefits of the study are obvious and successful. Figures and tables are very significantly helpful and comprehensible throughout the text. Moreover, the final results are quite optimistic for further use and research according to the writers.

Although, there are two minor comments that want some attention:

  1. Some bracketing without closing are in need of attention: in page 5 and line 15 “studies (for the list, see…”, in page 9 and line 19 “a) systematic processing (ES =…”.
  2. For the bibliographical references: 34 in page 6, and all in sections 2.3. RISP Constructs, 2.4. Protective Behaviors and 2.5. Analyses, there is not enough context for them to be used that way. The studies that there are referring to are not always clear, and we have not enough information about their content to understand it. I had to read some of the articles to comprehend your point.
